# A Rotatable Paper Device Integrating Reverse Transcription Loop-Mediated Isothermal Amplification and a Food Dye for Colorimetric Detection of Infectious Pathogens

**DOI:** 10.3390/bios12070488

**Published:** 2022-07-04

**Authors:** Hanh An Nguyen, Heewon Choi, Nae Yoon Lee

**Affiliations:** Department of BioNano Technology, Gachon University, 1342 Seongnam-daero, Sujeong-gu, Seongnam-si 13120, Gyeonggi-do, Korea; hanhannguyen96@gmail.com (H.A.N.); heewonchoi928@gmail.com (H.C.)

**Keywords:** carmoisine, reverse transcription loop-mediated isothermal amplification (RT-LAMP), rotatable paper device, SARS-CoV-2, multiplex detection

## Abstract

In this study, we developed a rotatable paper device integrating loop-mediated isothermal amplification (RT-LAMP) and a novel naked-eye readout of the RT-LAMP results using a food additive, carmoisine, for infectious pathogen detection. Hydroxyl radicals created from the reaction between CuSO_4_ and H_2_O_2_ were used to decolor carmoisine, which is originally red. The decolorization of carmoisine can be interrupted in the presence of DNA amplicons produced by the RT-LAMP reaction due to how DNA competitively reacts with the hydroxyl radicals to maintain the red color of the solution. In the absence of the target DNA, carmoisine is decolored, owing to its reaction with hydroxyl radicals; thus, positive and negative samples can be easily differentiated based on the color change of the solution. A rotatable paper device was fabricated to integrate the RT-LAMP reaction with carmoisine-based colorimetric detection. The rotatable paper device was successfully used to detect SARS-CoV-2 and SARS-CoV within 70 min using the naked eye. *Enterococcus faecium* spiked in milk was detected using the rotatable paper device. The detection limits for the SARS-CoV-2 and SARS-CoV targets were both 10^3^ copies/µL. The rotatable paper device provides a portable and low-cost tool for detecting infectious pathogens in a resource-limited environment.

## 1. Introduction

The ongoing severe acute respiratory syndrome coronavirus 2 (SARS-CoV-2) pandemic has become an unprecedented worldwide challenge. Although SARS-CoV-2 shares high genetic similarity with SARS-CoV, which caused an epidemic in China in 2002, the infection rate of SARS-CoV-2 is much higher than SARS-CoV [1,2]. As reported by the World Health Organization, there have been more than 267 million confirmed cases and five million deaths worldwide to date. In the race to slow the spread of SARS-CoV-2, diagnostic tests have become one of the most important measures to effectively respond to the pandemic; however, many countries are struggling to adapt to the demand for SARS-CoV-2 testing due to the high infection rate of SARS-CoV-2, which exceeds the healthcare resources of less economically developed countries [3,4]. Scaling-up diagnostic testing capacity has become a global focus by providing testing outside laboratory settings and reducing resource burdens; thus, a low-cost, rapid, and user-friendly diagnostic test is required. Although reverse transcription polymerase chain reaction (RT-PCR) is the gold-standard nucleic acid amplification test (NAAT) for screening SARS-CoV-2, it requires 25 to 30 thermal cycles. In addition, the process requires trained personnel to conduct the reaction [5]; therefore, RT-PCR does not meet the demand for a decentralized system.

To overcome the limitations of RT-PCR, reverse transcription loop-mediated isothermal amplification (RT-LAMP) has been introduced. RT-LAMP is an isothermal technique that can be used to detect numerous types of viral pathogens, including SARS-CoV-2, at a constant temperature. The RT-LAMP reaction simultaneously uses reverse transcriptase and the *Bst* enzyme, which has strong strand displacement activity, to amplify the RNA targets for detecting viruses [6,7,8]. RT-LAMP has clear advantages over RT-PCR because RT-LAMP is performed at a single temperature and is, therefore, is suitable for point-of-care testing (POCT), which satisfies the requirements for a decentralized system for SARS-CoV-2 testing.

Numerous strategies have been described for monitoring DNA production after RT-LAMP reaction. In one strategy, colorimetric detection can be used to directly visualize RT-LAMP amplicons via the naked eye, without the need for equipment. Phenol red, which is a pH indicator, is one of the most common dyes used for colorimetric detection. Phenol red can detect RT-LAMP amplicons by sensing the significant decrease in pH from alkaline to acidic. RT-LAMP coupled with phenol red for naked-eye detection has potential as a decentralized system for pathogen detection [9,10]. The buffer used in the RT-LAMP reaction plays a significant role in the color change, and the concentration of the buffer requires precise control; thus, the buffer concentration should be decreased. This, in turn, decreases the efficiency of the RT-LAMP reaction [11]. Alternatively, SYBR Green I, a fluorescent pH-independent dye, has been used to monitor RT-LAMP reactions. Although SYBR Green I is not affected by the reaction buffer, it requires an expensive ultraviolet transilluminator to visualize the results and the toxicity of SYBR Green I is relatively high [12]; therefore, a new dye that can overcome the limitations of current dyes for on-site visualization of RT-LAMP amplicons would encourage a wider acceptance from general users.

To take advantage of previously developed methods to display the RT-LAMP results, we introduced a less toxic dye, carmoisine, for RT-LAMP monitoring. Carmoisine is used extensively as a food additive and has not demonstrated mutagenic or carcinogenic effects, unlike SYBR Green I [13]. Carmoisine is an azo dye composed of two naphthalene subunits joined by an azo linkage (–N=N–) [14,15]. Using hydroxyl radicals created from the reaction between CuSO_4_ and H_2_O_2_, the azo linkage of carmoisine can be degraded to form two naphthalene molecules. Through degradation of the azo linkage, the dye can be decolorized [16,17,18,19]; however, the presence of a large number of DNA molecules reduces the effect of hydroxyl radicals on carmoisine because DNA can competitively react with hydroxyl radicals [20,21]. In the absence of DNA amplicons, the color of carmoisine is reduced from red to light pink; in the presence of DNA amplicons, the color of carmoisine remains red (Figure 1).

Recently, paper devices have gained considerable attention owing to the potential for the rapid and low-cost diagnosis of pathogens. A paper device is a small analytical device that can integrate multiple processes required for nucleic acid amplification tests into paper materials. The paper device can be simply operated by folding, sliding, or rotating, which makes it attractive for decentralized systems as it can replace several bulky systems, such as thermal cyclers, gel electrophoresis, and ultraviolet transilluminators [22,23]. In this study, we fabricated a rotatable paper device to integrate RT-LAMP and carmoisine-based colorimetric detection for pathogen detection.

## 2. Materials and Methods

### 2.1. One-Step RT-LAMP Assay

A one-step RT-LAMP assay was performed to detect the RNA of SARS-CoV-2, SARS-CoV, and human coronavirus OC43 (HCoV-OC43). RNA samples of these viral pathogens were extracted and collected from infected patients by Vircell Company. All primer sets for the amplification of the E gene of SARS-CoV-2 and SARS-CoV and the N2 gene of HCoV-OC43 were designed using PrimerExplorer Version 5. The primer sequences are listed in Appendix A. The RT-LAMP solution (25 µL) contained 1× isothermal amplification buffer, 1 mM dNTP, 2 mM MgSO_4_, 3.2 µM of each inner primer (FIP and BIP), 0.4 µM of each outer primer (F3 and B3), 1.6 µM of each loop primer (LF or LB), 0.18 U/µL of WarmStart RTx Reverse Transcriptase, and 0.096 U/µL of *Bst* 2.0 WarmStart DNA Polymerase. The RT-LAMP reaction was performed at 42 °C for 15 min, followed by incubation at 65 °C for 45 min. To examine the reliability of the experiments, negative controls that did not contain a DNA template were used.

### 2.2. Carmoisine-Based Detection of the RT-LAMP Amplicons

For the detection of RT-LAMP amplicons, 2 mg/mL carmoisine, 100 mM copper (II) sulfate pentahydrate, 3.5% H_2_O_2_, and RT-LAMP amplicons were mixed at a volume ratio of 3:1:1:5. The mixture was incubated at room temperature for 10 min. The presence and absence of RT-LAMP products corresponding to the positive and negative samples were directly observed via the naked eye. Gel electrophoresis was performed to validate the results. The color intensity of carmoisine was analyzed using ImageJ software.

### 2.3. Fabrication of the Rotatable Paper Device

A rotatable paper device was fabricated to integrate the RT-LAMP with colorimetric detection. The overall structure of the rotatable paper device is shown in Figure 2. The rotatable paper device consists of a rotatable part stacked on a poly(methyl methacrylate) (PMMA) substrate containing reaction chambers. To pattern the reaction chambers on the PMMA substrate, AutoCAD was used. From there, a computer numerical control (CNC) machine was used to cut the rectangular PMMA substrate (53 mm × 62.5 mm, 2 mm thickness) and engraved eight reaction chambers (4 mm diameter, 1.8 mm thickness). The chambers were arranged radially on the PMMA substrate. The rotatable part consisted of an upper piece of cardstock paper (53 mm × 62.5 mm), a thick and durable type of paper, that was placed on a convex rotator (49 mm diameter) and aligned with a lower concave piece of cardstock paper (53 mm × 62.5 mm). The rotator was fabricated by coating the cardstock paper with PDMS to create a waterproof surface. Eight holes (4 mm diameter) were radially arranged on the upper cardstock paper and the rotator. Eight reaction chambers on the PMMA substrate, eight holes on the upper cardstock paper, and the rotator were vertically aligned in the same position. Eight RT-LAMP-fused Whatman paper discs (4 mm diameter) were placed inside the reaction chambers. In addition, eight Whatman paper discs (4 mm diameter) were radially placed on top of the cardstock paper for colorimetric detection. The reaction chambers were treated with 1% bovine serum albumin (BSA) to prevent adsorption of RT-LAMP reagents onto the PMMA surface. The structure of the rotatable paper device allowed it to switch from an open to a closed state via rotation.

To create waterproof paper, the PDMS coating process was performed according to our previous study [24]. First, the PDMS prepolymer and curing agent were thoroughly mixed at a ratio of 10:1 (*w*/*w*). Following this step, air bubbles inside the mixture were removed by placing the mixture in a vacuum oven for 20 min. The mixture was then brushed onto the surface of the cardstock paper. Finally, the PDMS-coated paper was incubated at 80 °C for 6 h.

### 2.4. Operation of the Rotatable Paper Device

The entire operation of the rotatable paper device is shown in Figure 3. First, 1 µL of viral RNA was introduced into the reaction chambers containing RT-LAMP-fused paper discs. Next, 15 µL of sterile water was added to each reaction chamber, allowing RT-LAMP to occur. The rotatable paper device was then closed by rotating the rotator, followed by heating at 42 °C for 15 min and incubating at 65 °C for 45 min on a hot plate. To discriminate between the negative and positive samples, 3 µL carmoisine (2 mg/mL), 1 µL CuSO_4_ (100 mM), and 1 µL H_2_O_2_ (3.5%) with a volume ratio of 3:1:1, respectively, were added to the reaction chambers. The mixture was incubated at room temperature for 10 min. After 10 min, 2 µL of the solution in the reaction chambers was loaded onto the Whatman paper discs attached on the upper cardstock paper for naked-eye observation of the results.

### 2.5. Sensitivity of Carmoisine-Based Detection Performed Inside Tubes

The sensitivity of carmoisine-based detection was verified by conducting ten-fold serial dilutions of RNA from SARS-CoV-2, SARS-CoV, and HCoV-OC43. RNA templates were incubated at 42 °C for 15 min, followed by another incubation at 65 °C for 45 min. The presence of RT-LAMP amplicons was verified by the addition of carmoisine. The results were validated using agarose gel electrophoresis. The gel was photographed under UV light using an imaging system, Bio-Rad Molecular Imager Gel ChemiDoc XR.

### 2.6. Multiplex Detection Using the Rotatable Paper Device

To evaluate the potential of the rotatable paper device for multiplex detection, RNA from SARS-CoV-2 and SARS-CoV was mixed and loaded into the reaction chambers containing RT-LAMP reagent-soaked paper discs. Each paper disc was soaked with different primer sets to amplify different targets, including SARS-CoV-2, SARS-CoV, and HCoV-OC43. Negative and positive controls were used to prevent false negatives and false positives. For verification of the procedure, 2 µL of the amplicons were used for agarose gel electrophoresis.

## 3. Results and Discussions

### 3.1. Colorimetric Detection of RT-LAMP Amplicons

Figure 4a,b show the results of the colorimetric detection of viral RNA templates using RT-LAMP and carmoisine-based colorimetric detection. CuSO_4_ (100 mM) and H_2_O_2_ (3.5%) were used to generate hydroxyl radicals which could decolorize azo dyes, particularly carmoisine. The decolorization of carmoisine occurs slowly when hydroxyl radicals react with DNA amplicons; therefore, only samples containing DNA amplicons remained red, while the color of carmoisine decreased to light pink in the absence of DNA amplicons. Among the three common pathways for the reaction of hydroxyl radicals including electron abstraction, hydrogen abstraction, and double bond addition, hydroxyl radicals have been reported to react with DNA through hydrogen abstraction from DNA molecules to hydroxyl radicals and the addition of hydroxyl radicals to the –C=C– bonds on DNA strands [20,21,25]; however, hydroxyl radicals react not only with DNA molecules, but also with other macromolecules such as proteins [26,27,28]. The addition of BSA, which is a type of protein used for surface passivation, can interfere with colorimetric detection. To evaluate the resistance of carmoisine-based detection to proteins, a range of concentrations of BSA from 0% to 1% was added to the RT-LAMP reagents, followed by the addition of all agents required for carmoisine-based detection. As a result, BSA concentration ranging from 0% to 1% did not interfere with carmoisine-based detection.

Although a positive reaction can remain red after 10 min, as previously mentioned, the red color observed in the positive reaction could gradually decolor and become colorless. As shown in Appendix A, after 20 min at room temperature, the negative and positive samples became colorless because the excess hydroxyl radicals kept producing decolorized carmoisine; therefore, after the colorimetric reaction, the mixture containing the RT-LAMP products, carmoisine, CuSO_4_, and H_2_O_2_ was taken out and soaked on a Whatman paper disc to stop the effect of the hydroxyl radicals on carmoisine.

Although various colorimetric techniques for monitoring RT-LAMP results have been developed, there are limitations. For example, numerous studies have reported the ability of gold nanoparticles for RT-LAMP and LAMP monitoring [29,30,31], but the complicated and strict preparation process for gold nanoparticles requires many steps, hindering their application in low-resource settings [32]. Camoisine-based detection does not require a strict preparation process, unlike gold nanoparticle-based methods; therefore, carmoisine-based detection has high potential as a pathogen detecting method in low-resource settings.

### 3.2. Influence of the Reagents Addition Sequence on Colorimetric Detection

As shown in Figure 4c, carmoisine-based colorimetric detection of RT-LAMP amplicons was affected by the order of adding carmoisine, CuSO_4_, and H_2_O_2_. Strong decolorization of the carmoisine occurred when carmoisine was added before the generation of hydroxyl radicals—that is, before the simultaneous addition of CuSO_4_ and H_2_O_2_. When the hydroxyl radicals are produced before the addition of carmoisine, the hydroxyl radicals react quickly with other substances, such as primers, enzymes, and MgSO_4_; thus, a decrease in the concentration of hydroxyl radicals could reduce the decolorization of carmoisine. In addition, the clearest difference between the negative and positive samples was observed when carmoisine was added, followed by CuSO_4_ and H_2_O_2_.

### 3.3. Effect of Heat Incubation Time of RT-LAMP on Pathogen Detection

To evaluate the effect of the heat incubation time of the RT-LAMP reaction on colorimetric detection, RNA from SARS-CoV-2 was reverse transcribed to cDNA, followed by further incubation at 65 °C for various times ranging from 0 min to 45 min for subsequent cDNA amplification. As shown in Figure 5a, DNA amplicons were detected after heat incubation for 30–45 min. The strongest intensity of ladder-like patterns produced by gel electrophoresis was observed at an incubation time of 45 min. Similarly, the strongest red color was observed when cDNA amplification was performed for 45 min. Based on these results, a heat incubation time of 45 min was selected for cDNA amplification.

### 3.4. Specificity Test

For most diagnostic tests, the ability to discriminate SARS-CoV-2 from other closely related coronaviruses is vital and clinically significant; therefore, RNA from other coronaviruses, such as SARS-CoV and HCoV-OC43, were used to demonstrate the specificity of SARS-CoV-2 detection. A bacteria strain belonging to the human pathogen, *Enterococcus faecium* (*E. faecium*), was also used. We used a primer set for SARS-CoV-2 amplification to detect SARS-CoV, HCoV-OC43, and *E. faecium*. Figure 5b shows the results of this specificity test. From lane 2 to 5, SARS-CoV-2 primers were added into samples containing SARS-CoV, HCoV-OC43, and *E. faecium* nucleic acid, respectively. Lane 1 can be considered a negative control because RT-LAMP was performed without the RNA template. Cross-reactivity with other viruses or bacteria was not observed; thus, the SARS-CoV-2 primer set only initiated RT-LAMP reaction in the presence of SARS-CoV-2 templates. Negative controls were used to validate the results. The graph on Figure 5b shows the mean gray values obtained from the results of colorimetric detection. Only sample 5, representing the presence of SARS-CoV-2 target, showed significantly higher mean gray values than the others. The specificity of primer sets for SARS-CoV and HCoV-OC43 detection are shown in Appendix A. The three primer sets for detecting SARS-CoV-2, SARS-CoV, and HCoV-OC43 used in this study were highly specific.

### 3.5. Sensitivity of RT-LAMP Coupled with Carmoisine-Based Detection

The sensitivity of the RT-LAMP reaction coupled with carmoisine-based detection was determined using ten-fold serial dilution of RNA. RNA samples from three respiratory viruses (SARS-CoV-2, SARS-CoV, and HCoV-OC43) with concentrations ranging from 10^0^ to 10^4^ copies/µL were loaded into the tubes. The RT-LAMP reaction was conducted at 42 °C for 15 min followed by incubation at 65 °C for 45 min. After the RT-LAMP reaction, carmoisine, CuSO_4_, and H_2_O_2_ were added sequentially and the tubes were incubated at room temperature. A negative control was used to check for false positive results. An intense red color was observed when samples containing 10^3^ and 10^4^ copies/µL of RNA from SARS-CoV-2 were used (Figure 6a). The red color of carmoisine indicated that samples containing 10^3^ and 10^4^ copies/µL of RNA were successfully amplified and detected. Conversely, paper discs displaying a light pink color were observed for samples containing RNA with concentrations ranging from 10^0^ to 10^2^ copies/µL, indicating that the RT-LAMP reaction was not successfully amplified. For gel electrophoresis results, ladder-like bands were observed when the samples contained 10^3^ or 10^4^ copies/µL. Similarly, sensitivity tests were performed using HCoV-OC43 and SARS-CoV targets (Figure 6b,c). The combination of RT-LAMP and colorimetric detection could also be used to detect RNA via the naked eye from HCoV-OC43 and SARS-CoV, with the concentration ranging from 10^3^ to 10^4^ copies/µL; concurrently, the concentrations of RNA from both HCoV-OC43 and SARS-CoV, ranging from 10^0^ to 10^3^ copies/µL were not detected by colorimetric detection and gel electrophoresis.

### 3.6. Reproducibility Test for SARS-CoV-2 Detection Using the Rotatable Paper Device

To determine the reproducibility of the rotatable paper device, 1 µL of RNA from SARS-CoV-2 and 15 µL of sterile water were loaded into the reaction chambers on the device. The rotatable paper device was then closed by rotating the rotator of the device. The RT-LAMP reagents stored inside the paper discs stimulated the reverse transcription to generate cDNA by heat incubation at 42 °C for 15 min. The cDNA was continuously amplified by further incubation at 65 °C for 45 min. The RT-LAMP amplicons were visualized as shown in Figure 7. A strong red color was observed on paper discs 5–8, indicating the presence of SARS-CoV-2 RNA; conversely, paper disc 14 was light pink owing to the absence of SARS-CoV-2 RNA. The mean gray values from samples 5–8 were also significantly higher than those of samples 1–4. To validate the results, 1 µL of RT-LAMP amplicons amplified using the rotatable paper device was used for gel electrophoresis. Similarly, intense ladder-like patterns were observed in lanes 5–8, whereas ladder-like patterns were not observed in lanes 1–4, corresponding to the presence and absence of SARS-CoV-2 RNA, respectively.

### 3.7. Multiplex Detection Using Rotatable Paper Device

For most diagnostic tests, the ability to distinguish multiple pathogens that cause similar symptoms can enhance operational efficiency and improve cost-effectiveness [33]; therefore, we further estimated the potential for multiple-pathogen detection using the rotatable paper device (Figure 8). The mixture of SARS-CoV and SARS-CoV-2 RNA was loaded into chambers 6–8, where the primer sets for amplifying SARS-CoV, HCoV-OC43, and SARS-CoV-2, respectively, were contained. As a result, both SARS-CoV-2 and SARS-CoV were detected via the naked eye using the rotatable paper device. Negative controls were used to prevent false positives, and positive controls were used to prevent false negatives. The mean gray values obtained from the positive controls (paper discs 3–5) and paper discs 6 and 8, corresponding to the presence of the target RNA templates, were significantly higher than those of the negative controls (paper discs 1 and 2) and paper disc 7, corresponding to the presence of the non-matched RNA template. This rotatable paper device can extend its application to detect various other pathogens by redesigning the primers and loading paper discs soaked with different primers.

### 3.8. Real Sample Analysis

To evaluate the feasibility and application perspective in real samples, the rotatable paper device was used to detect *E. faecium* spiked in milk. The spiked milk samples were heated at 95 °C for 30 min to break the bacterial membrane and expose bacterial DNA. Subsequently, the exposed DNA was amplified and detected using the rotatable paper device. Unspiked milk samples were used as negative controls. Figure 9 shows the results of real sample analysis using the rotatable device. Red color was observed on paper discs 2, 4, and 6, indicating the presence of *E. faecium* in the milk samples. The amplified DNA of *E. faecium* in chambers 2, 4, and 6 competitively reacted with hydroxyl radicals, which resulted in the maintenance of red color of carmoisine. Conversely, paper discs 1, 3, and 5 displayed a light pink color owing to the absence of *E. faecium* DNA. Hydroxyl radicals strongly reacted with carmoisine in the absence of DNA amplicons, which resulted in the decolorization of carmoisine. Gel electrophoresis also validated the result obtained from the rotatable paper device.

## 4. Conclusions

We have successfully developed a rotatable paper device to integrate RT-LAMP with carmoisine-based colorimetric strategy for multiplex pathogen detection. The obtained visual outcomes demonstrated the colorimetric detection potential of RT-LAMP combined with carmoisine without the use of bulky equipment. In addition, the carmoisine-based detection method met the safety requirements for the identification of specific pathogens. The rotatable paper device is user friendly because it can be simply operated by rotation. RT-LAMP amplification, coupled with naked-eye detection using carmoisine, showed high specificity, sensitivity, and simplicity. The rotatable paper device was successfully used to simultaneously detect multiple pathogens, such as SARS-CoV-2 and SARS-CoV, within 70 min. The spiked milk sample with *E. faecium* was detected using the rotatable paper device. We believe that this rotatable paper device satisfies the demand for SARS-CoV-2 screening with limited resources to prevent the spread of infectious pathogens.

## Figures and Tables

**Figure 1 biosensors-12-00488-f001:**
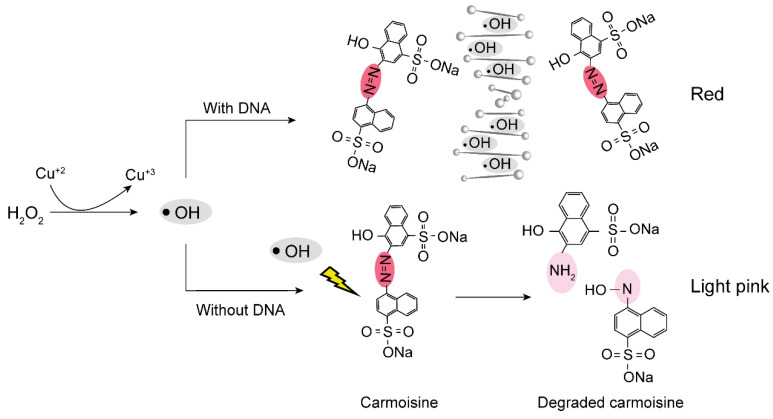
Schematic demonstrating the mechanism for the colorimetric detection of RT-LAMP amplicons using carmoisine.

**Figure 2 biosensors-12-00488-f002:**
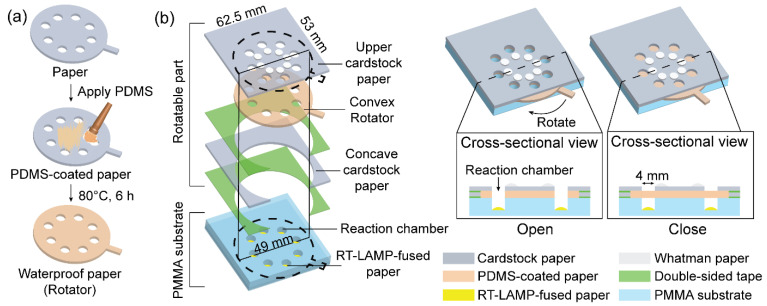
Schematic showing the fabrication of the rotatable paper device: (**a**) PDMS-coating process and (**b**) overall structure of rotatable paper device.

**Figure 3 biosensors-12-00488-f003:**
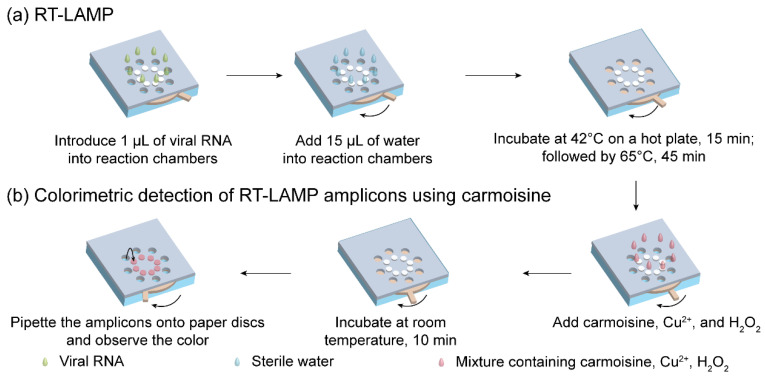
Operation of the rotatable paper device. Procedure for (**a**) RT-LAMP reaction and (**b**) colorimetric detection using carmoisine.

**Figure 4 biosensors-12-00488-f004:**
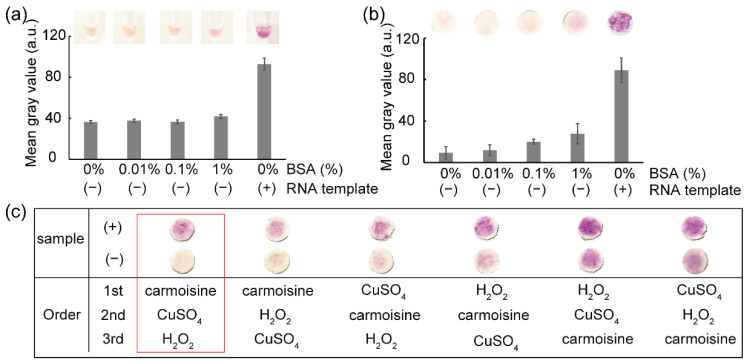
The colorimetric detection of pathogens. The results show the effect of BSA for (**a**) in-tube and (**b**) paper-based colorimetric detection: (−) RT-LAMP reaction not containing RNA template and (+) RT-LAMP reaction containing RNA template. Image (**c**) shows the effect of the sequence of the reagent’s addition in colorimetric detection.

**Figure 5 biosensors-12-00488-f005:**
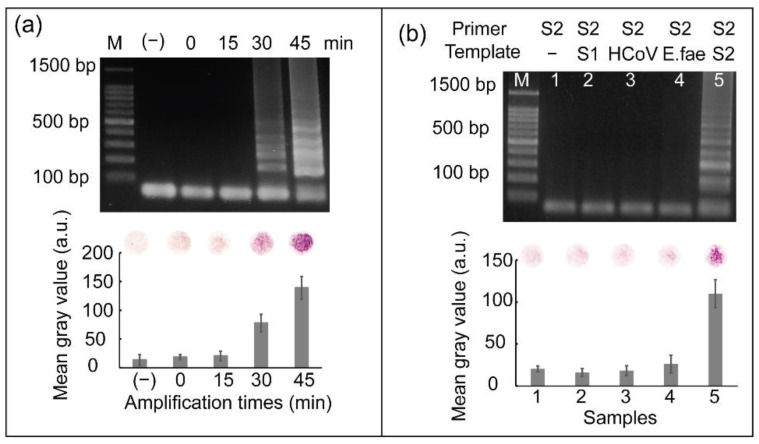
(**a**) Results showing the effect of amplification time on screening SARS-CoV-2 and (**b**) results of the specificity test of RT-LAMP coupled with colorimetric detection for screening SARS-CoV-2. Here, M represents 100 bp ladder, lane 1: RT-LAMP reaction not containing RNA template and lanes 2–5: RT-LAMP reaction containing RNA from SARS-CoV, HCoV-OC43, *E. faecium*, and SARS-CoV-2, respectively. All the experiments were repeated three times.

**Figure 6 biosensors-12-00488-f006:**
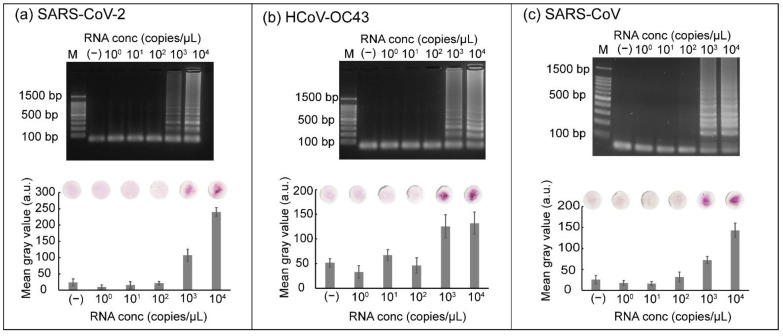
Results showing (**a**) the sensitivity of SARS-CoV-2 detection mediated by carmoisine, (**b**) the sensitivity of HCoV-OC43 detection, and (**c**) the sensitivity of SARS-CoV detection. M represents 100 bp ladder. RNA concentrations ranging from 10^0^ to 10^4^ copies/µL were used. All experiments were repeated three times.

**Figure 7 biosensors-12-00488-f007:**
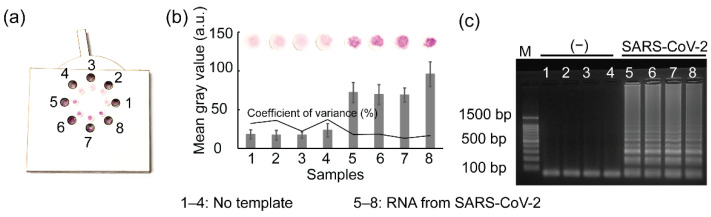
Results showing (**a**) SARS-CoV-2 detection using the rotatable paper device, (**b**) mean gray analysis, and (**c**) agarose gel electrophoresis. Lane M shows 100 bp ladder. Lanes 1–4 show RT-LAMP performance without RNA template. Lanes 5–8 show RNA amplification from SARS-CoV-2.

**Figure 8 biosensors-12-00488-f008:**
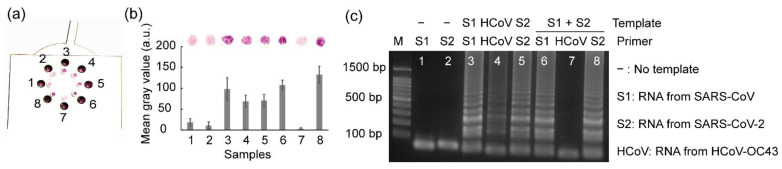
Results showing (**a**) multiplex detection using the rotatable paper device, (**b**) mean gray analysis, and (**c**) agarose gel electrophoresis. Lanes 1 and 2 show the negative controls. Lanes 3–5 show positive controls. Lanes 6–8 show multiplex detection. All the experiments were repeated three times.

**Figure 9 biosensors-12-00488-f009:**
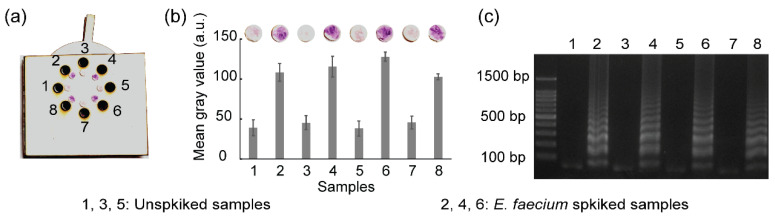
Results showing (**a**) the detection of *E. faecium* spiked in milk using the rotatable device, (**b**) mean gray analysis, and (**c**) agarose gel electrophoresis. All the experiments were repeated three times.

## Data Availability

Not applicable.

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
