# Peer review of "A Rotatable Paper Device Integrating Reverse Transcription Loop-Mediated Isothermal Amplification and a Food Dye for Colorimetric Detection of Infectious Pathogens"

_biosensors, 2022, doi:10.3390/bios12070488_

Round 1

Reviewer 1 Report

This manuscript (biosensors-1762928) reported a rotatable paper device integrating with loop-mediated isothermal amplification (RT-LAMP) for colormetric and visual detection of infectious pathogen by using carmoisine, which showed tremendous advantages for detecting infectious pathogens in a resource-limited environment. In gereral, this paper is an interesting and systematic work with comprehensive data supporting. After careful evaluations, I think it can be published in Biosensors after some minor revisions.

Detailed Comments:

1. Please carefully check and polish the language, there are some usage/grammatical issues.

2. Normally, CuSO4 and H2O2 were used to generate hydroxyl radicals by the Fenton-like reaction. However, the verification of hydroxyl radicals was not mentioned in the manuscript. Please provide relevant experiments (the oxidation of TMB, terephthalic acid, and electron paramagnetic resonance (EPR)) to confirm the existence of hydroxyl radicals.

3. Confusingly, the effect of the individual CuSO4 and H2O2 on the decolorization of carmoisine should be supplied in the manuscript.

4. Compared with previous colorimetric techniques for monitoring RT-LAMP, what are the innovations and advantages of this study?

5. How does the visual results translate into mean gray value? Please provide some explanation.

6. Please compare the detection sensitivity of SARS-CoV-2, SARS-CoV, and HCoV-OC43 detection in this study with previous studies.

7. The conclusion needs to be condensed and improved.

8. There are some writing errors in the references, the author need to check them totally.

Author Response

We would like to thank the reviewer for the helpful comments and suggestions. We have improved the manuscript following your comments.

Reviewer 2 Report

The authors present a very interesting new paper device for the amplification and detection of viruses and bacteria that can be also applied to any RNA and DNA sequence. The manuscript is well written and the experiments were appropriately designed. However, there are some points that have to be addressed in order to ensure the good performane of this device and to increase the impact of the manuscript.

- I believe that the term rotatable is not suitable for this device, as a piece of PDMS-coated paper is used to close and open the reactive chambers of the device by rotation. So, I suggest to remove this term from the manuscript.

- The authors report that this device is also suitable for multiplex detection. I believe that the most appropriate term here is the 'screening', because the authors aschieved to detect if there is any of the analyzed viruses or bacteria in the sample, but the cannot identify which one. Separate amplification reaction should then be then performed in order to identify the pathogen. So, please correct this throughout the manuscript.

- In Fig. 2 the authors should also explain how the device is heated. 

- at p. 3, line 108, please add the diluent of carmoisine.

- Figure 4a and b, please explain why in the positive sample no BSA is added.

- at paragraph 3.5, please replace the term 'senistivity' with the correct term of detectability.

- at paragraph 3.6, please calculate and include the %CVs by the grey values in order to estimate the reproducibility of the method. Reproducibility has to be determined for low, medium and high concentrations of all analytes.

- The detectability of E. faesium should also be determined. Please also add to the materials the primers used for the amplification of E. faesium.

- Figure S2, the specificity study here is not clear. Please explain in more details how the experiments were performed and explain better the results in this figure.

- Finally, the authors should compare their method with the existing ones for the analysis of all targets, eg. real time PCR, especially for the limit of detection that is very crucial for the detection of infection. The authors should also present more clearly the advantages of their method compared to the existing ones. 

Author Response

(The authors gave the same response as above.)

Reviewer 3 Report

In this paper the authors developed a rotatable paper device for infectious pathogen detection. They integrated in the rotatable paper device, a loop-mediated isothermal amplification (RT-LAMP) and a naked-eye readout of the RT-LAMP product using carmoisine (a food additive). The device was successfully used to detect SARS-CoV-2, SARS-CoV, within 70 min using the naked eye; and Enterococcus faecium spiked in milk. Overall, the rotatable paper device shows good performance with an easy to read and portable device. The work is well-done, with relevant experiments and controls. Also, the manuscript is well-written and is easy to follow.

One of the main novelties of this report is the use of carmoisine as indicator to detect the presence of RT-LAMP product. As the authors discussed in the manuscript, there are other indicators that are frecuntly used as indicators, such as pH indicators. I think it is critical that the author compare the analytical performance of carmoisine with a frequently used pH indicator. For instance, recently Dao Thi, V. L. et al (DOI: 10.1126/scitranslmed.abc7075) have shown a similar colorimetric detection of RT-LAMP product with a detection limit one or two orders of magnitude lower than the reported here. I suggested performing the experiments with the rotatable paper device the authors developed and a pH indicator frequently used for the colorimetric detection of the RT-LAMP product, to compare the results. Moreover, a more scholarly comparison of the sensor performance with similar reports in the literature has to be done. This discussion will be extremely valuable for the readers and will clarify the advantage of using carmoisine as an indicator.

Minor revisions:

1.       It seems that figure 1 has been cut (H2O2 is not complete). Also, it seems that the two presented structures of carmoisine are different . Aternatively the one after “without DNA” may be the product of the reaction with OH. It is not clear how to interpret the differences. Please, clarify.

2.       In the experimental section and Figure 3 it seems that carmoisine, CuSO4 and H2O2 are added as a mixture together, but reading the manuscript and looking at Figure 4c it seems that they are adding one after the other. Please, clarify.

3.       The concentration of CuSO4 and H2O2 are always maintained constant. Why are those concentrations chosen? Does it come from a previous work? Does the sensitivity and time of the experiments depend on those concentrations? Please, discuss or add references.

4.       Page 6, line 205. The authors compare the price of the assay they develop with a commercially available master mix. If the authors want to compare prices, a broader discussion and comparison need to be done. If not, the conclusion of this discussion is based on a simple comparison with one kit.

5.   Page 9, line 289. Reproducibility experiment. Please clarify in the text which samples are positive and which ones are negative.

6.   Please clarify what electron and hydrogen abstraction means

Author Response

(The authors gave the same response as above.)

Round 2

Reviewer 2 Report

The authors have respoded properly to most of the reviewers' suggestions.

However there are still to points that have to be corrected.

- The term sensitivity has to be replaced by the detectability. There are completely different terms. The authors here present the detectability and not the sensitivity of the method.

- A detailed comparison of their method with existing methods including the major analytical parameters, such as limit of detection, dynamic range, analysis time, real sample analysis, multiplicity, has to be included. Thus the authors have to present more clearly the advantages of their method compared to the existing methods.

Author Response

Thank you for the valuable comments.
